# Multicentre, national, investigator-initiated, randomised, parallel-group, register-based superiority trial to compare extended ECG monitoring versus standard ECG monitoring in elderly patients with ischaemic stroke or transient ischaemic attack and the effect on stroke, death and intracerebral bleeding: the AF SPICE protocol

Johan Engdahl [1], Kajsa Straat,[1] Eva Isaksson [1], Elisabeth Rooth,[1] Emma Svennberg,[2] Bo Norrving,[3] Mia von Euler,[4] Kjersti Hellqvist,[5] Weigang Gu,[6] Jakob O Ström,[4] Sara Själander,[7] Marie Eriksson,[8] Signild Åsberg,[9] Per Wester[1,7]

SÅ and PW are joint senior authors.

For numbered affiliations see end of article.

**Correspondence to**
Dr Johan Engdahl;
johan.engdahl@
regionstockholm.se

## ABSTRACT

**Introduction** Atrial fibrillation (AF) is a major risk factor for ischaemic stroke and transient ischaemic attack (TIA), and AF detection can be challenged by asymptomatic and paroxysmal presentation. Long-term ECG monitoring after ischaemic stroke or TIA is recommended by all major societies in cardiology and cerebrovascular medicine as a secondary prophylactic measure. However, data on stroke reduction are lacking, and the recommendations show significant diversity.

**Methods and analysis** AF SPICE is a multicentre, national, investigator-initiated, randomised, parallel-group, register-based trial comparing extended ECG monitoring versus standard ECG monitoring in patients admitted with ischaemic stroke or TIA, with a composite endpoint of stroke, all-cause-mortality and intracerebral bleeding. Patients aged ≥70 years without previous AF will be randomised 1:1 to control (standard ECG monitoring) or intervention (extended ECG monitoring). In the control arm, patients will undergo 48±24 hours (ie, a range of 24–72 hours) of continuous ECG monitoring according to national recommendations. In the intervention arm, patients will undergo 14+14 days of continuous ECG monitoring 3 months apart using an ECG patch device, which will provide an easy-accessed, well-tolerated 14-day continuous ECG recording. All ECG patch recordings will be read in a core facility. In cases of AF detection, oral anticoagulation will be recommended if not contraindicated. A pilot phase has been concluded in 2022, which will transcend into the main trial during 2023–2026, including approximately 30 stroke units. The sample size was calculated to be 3262 patients. The primary outcome will be collected from register data during a 36-month follow-up.

## STRENGTHS AND LIMITATIONS OF THIS STUDY

⇒ First register-based randomised clinical trial on secondary preventive ECG screening poststroke/transient ischaemic attack with stroke and all-cause mortality endpoints.
⇒ Participants were included from a national network of stroke units and central adjudication of ECG recordings in the intervention arm.
⇒ Robust inclusion criteria mirror the clinical real-world population; age at least 70 years for inclusion.
⇒ The Swedish stroke register (Riksstroke) is used for the provision of the majority of baseline data.
⇒ Endpoints from healthcare registers will not be further adjudicated.

**Ethics and dissemination** Ethical approval has been provided by the Swedish Ethical Review Authority, reference 2021–02770. The trial will be conducted according to the ethical principles of the Declaration of Helsinki and national regulatory standards. Positive results from the study have the potential for rapid dissemination in clinical practice.

**Trial registration number** NCT05134454.

## INTRODUCTION

Ischaemic stroke is one of the leading causes of mortality worldwide and a major cause of permanent disability in adults.[1] Despite the implementation of novel treatment strategies, morbidity and mortality remain significant.

Modifiable risk factors for stroke include hypertension, atrial fibrillation (AF), hyperlipidaemia, smoking and diabetes.[2 3] Several of these risk factors have been undetected for a long time since they remain asymptomatic.

During 2018, the stroke incidence in Sweden was 223 per 100 000 in men and 191 per 100 000 in women.[4] At the 3-month follow-up poststroke, 26% were deceased or dependent on activities of daily living.[4 5] The risk for stroke recurrence is highest during the first 3–6 months following admission, and the 1-year recurrence risk has been reported to be 3–5%.[6 7]

The societal burden of the disease is significant. Disability with a need for long-term care, and in younger patients, impaired work ability entails enormous societal costs.[8] Strokes due to AF are more severe and have significantly higher mortality than other stroke cases.[9]

AF is the most prevalent clinical arrhythmia, with a steeply increasing incidence with advancing age.[10] AF is also one of the strongest risk factors for stroke.[3] However, the increased stroke risk associated with AF can be markedly reduced by oral anticoagulation (OAC) treatment.[11] Unfortunately, AF is paroxysmal and asymptomatic in a significant proportion of stroke patients, leading to lower detection rates and frequently leaving stroke survivors with an undiagnosed and untreated risk factor and a higher risk of stroke recurrence.[12] Despite this, no study so far has reported evidence of the benefit of ECG AF screening in terms of reduced stroke recurrence and mortality after a stroke event.

Randomised controlled trials[13] (CRYSTAL-AF[14] and EMBRACE[12]) have shown that prolonged ECG monitoring compared with standard of care (at the time of 24-hour ECG) increased detection of AF in patients with cryptogenic stroke.

The FIND-AF$_{randomised}$ trial[15] also showed increased detection of AF and included a broader group of participants, suggesting that prolonged ECG monitoring should be used in all patients with ischaemic stroke.

The MonDAFIS study[16] was powered to evaluate the proportion of anticoagulation therapy 12 months after index stroke. The standard of care was compared with prolonged inhospital ECG monitoring (maximum 7 days). AF detection was increased in the intervention group, but this had no significant effect on the rate of anticoagulation therapy at 12 months.

AF screening after an ischaemic stroke event is currently recommended by major international societies (table 1).[17–23] The recommendations vary markedly with respect to the method of screening and duration of screening. There is, however, a body of data on the proportion of AF detection using different monitoring strategies.[24]

The development of ECG modalities and clinical practice for AF screening postischaemic stroke and TIA have evolved towards longer and more resource-demanding investigations, as noted in table 1, despite the lack of data on clinical endpoints like stroke. Diversity within recommendations has promoted unequal care even within countries and regions. With this background, there is an urgent need for a randomised trial to investigate the efficacy of secondary prophylactic AF screening following ischaemic stroke.

## Choice of comparators

In some of the recommendations on AF screening for postischaemic stroke from international societies, a basic monitoring duration of 24 hours of continuous ECG is recommended.[18 20] In the recommendations from the Swedish Board of Health and Welfare, continuous ECG monitoring of 24–48 hours is recommended as the baseline investigation for patients with ischaemic stroke patients, and a large share of stroke units in Sweden adhere to this recommendation.

## METHODS AND ANALYSIS

### Primary objectives

To determine if extended continuous ECG screening (14+14 days) is superior to standard ECG screening (48±24 hours, that is, range 24–72 hours) in reducing the risk of the combined endpoint of ischaemic stroke, intracerebral bleeding and all-cause mortality in elderly patients admitted to the hospital for ischaemic stroke or TIA.

### Key secondary objectives

The key secondary objectives are to estimate the treatment effect on the primary outcome and determine if there are any differences between extended and standard ECG screening for:
- Ischaemic stroke.
- All-cause mortality
- Intracranial bleeding
- Major bleeding.
- Myocardial infarction.
- Pacemaker implantation.
- Cost-effectiveness.

### Other secondary objectives
- To study the association between age and AF detection.
- To study the proportion of other relevant arrhythmias detected at ECG screening.
- To study predictors for AF detection.
- To study signal time and signal quality achieved at ECG screening.
- To study long-term adherence to OAC treatment.
- To study the feasibility of patients' application of the second ECG patch.
- To study patient reported experience, collected 3 months after inclusion in the main trial.
- Per protocol analysis of the primary outcome.
- Win ratio analysis.

### Trial design

AF SPICE is designed as a multicentre, national, investigator-initiated, randomised, parallel-group, register-based superiority trial to compare extended ECG

**Table 1**  ECG screening poststroke society recommendations

| Society | Recommendation summary | Year | Ref |
|---|---|---|---|
| European Stroke Organisation | In adult patients with ischaemic stroke or TIA of undetermined origin, we recommend a prolonged cardiac monitoring instead of standard 24 hours monitoring to increase the detection of subclinical AF.<br>In adult patients with ischaemic stroke or TIA of undetermined origin, we recommend longer duration of cardiac rhythm monitoring of more than 48 hours and if feasible with n implantable loop recorder to increase the detection of subclinical AF. | 2022 | [21] |
| American Heart Association/ American Stroke Association | In patients suspected of having a stroke or TIA, an ECG is recommended to screen for AF and atrial flutter and to assess for other concomitant cardiac conditions.<br>In patients with cryptogenic stroke who do not have a contraindication to anticoagulation, long-term rhythm monitoring with mobile cardiac outpatient telemetry, an implantable loop recorder or other approach is reasonable to detect intermittent AF. | 2021 | [19] |
| European Society of Cardiology | In patients with acute ischaemic stroke or TIA and without previously known AF, monitoring for AF is recommended using a short-term ECG recording for at least the first 24 hours, followed by continuous ECG monitoring for at least 72 hours whenever possible.<br>In selected stroke patients without previous known AF, additional monitoring using long-term non-invasive ECG monitors or insertable cardiac monitors should be considered to detect AF. | 2020 | [18] |
| Swedish board of health and welfare | Healthcare should provide a long-term ECG for 24–48 hours.<br>long-term ECG for more than 48-hour duration using Holter recording or inpatient telemetry ECG in selected cases. | 2020 | |
| NICE | Reveal LINQ is recommended as an option to help to detect AF after cryptogenic stroke only if: non-invasive ECG monitoring has been done and a cardiac arrhythmic cause of stroke is still suspected. | 2020 | [15] |
| Canadian Cardiovascular Society/ Canadian Heart Rhythm Society | We recommend at least 24 hours of ambulatory ECG monitoring to identify AF in patients with non-lacunar ESUS. We suggest additional monitoring for AF detection (eg, prolonged external loop recorder or implantable cardiac monitoring, where available) be performed for selected older patients with non-lacunar ESUS in whom AF is suspected but unproven. | 2020 | [16] |
| National Heart Foundation of Australia and the Cardiac Society of Australia and New Zealand | Screening for AF in patients with ESUS recommendation: for patients with ESUS, longer term ECG monitoring (external or implantable) should be used. | 2018 | [17] |
| Royal College of Physicians | People with ischaemic stroke or TIA who would be eligible for secondary prevention treatment for AF (anticoagulation or left atrial appendage device closure) should undergo a period of prolonged (at least 12 hours) cardiac monitoring. People with ischaemic stroke or TIA who would be eligible for secondary prevention treatment for AF and in whom no other cause of stroke has been found should be considered for more prolonged ECG monitoring (24 hours or longer), particularly if they have a pattern of cerebral ischaemia on brain imaging suggestive of cardioembolism. | 2016 | [20] |

AF, atrial fibrillation; ESUS, embolic stroke of uncertain source; TIA, transient ischaemic attack.

monitoring versus standard ECG monitoring in elderly patients admitted with ischaemic stroke or TIA with the combined endpoint of recurrent ischaemic stroke, all-cause mortality and intracerebral bleeding (figure 1).

**Study setting**

AF SPICE will recruit participants among admissions for ischaemic stroke or TIA in stroke units in Swedish hospitals. Inclusion is also possible from non-stroke units given that the patient is admitted for ischaemic stroke or TIA and identified by the site's local investigator or delegate.

A pilot trial commenced in January 2022 and is now concluded, converting into the main study in 2023. The pilot trial evaluated inclusion rates, feasibility and

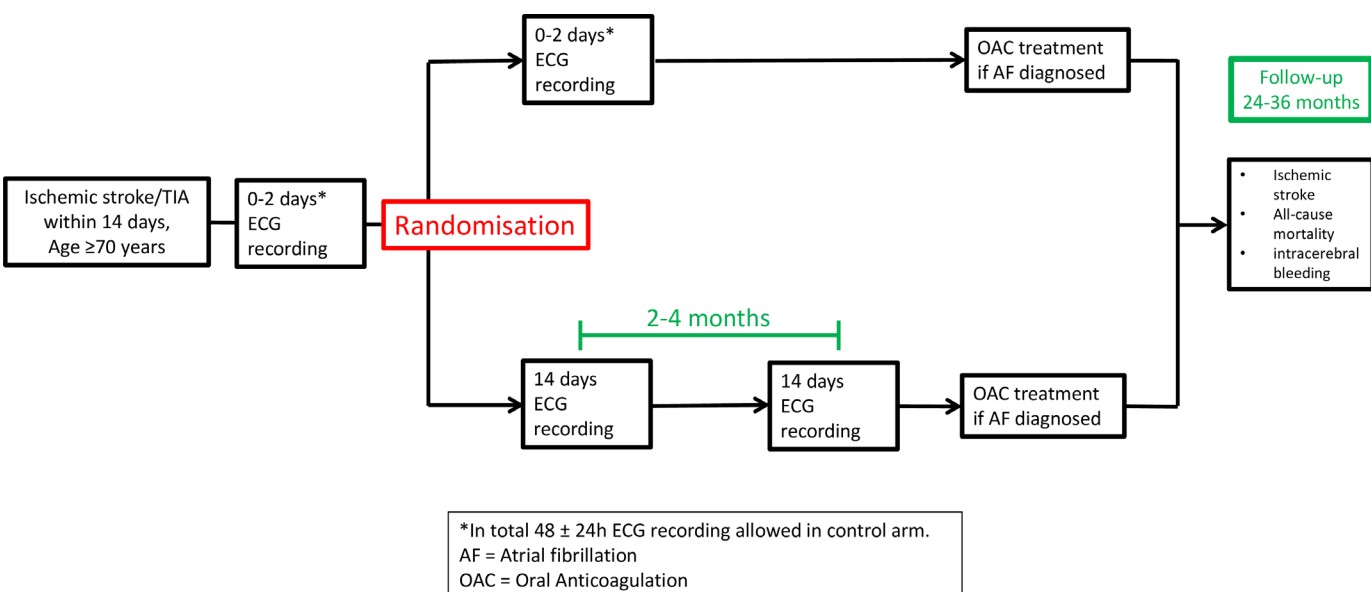

**Figure 1** Overview of study design, AF SPICE.

adherence to ECG recorders and the database with a randomisation system.

## Eligibility criteria
### Inclusion criteria

1. Patients aged ≥70 years admitted to a stroke unit with a diagnosis of ischaemic stroke or TIA within 14 days of inclusion. Stroke is clinically defined as the acute onset of a focal neurological deficit of presumed vascular origin lasting for ≥24 hours or resulting in death. Stroke is further categorised as ischaemic or as intracerebral or subarachnoid haemorrhage based on CT and/or MRI of the brain. TIA is defined as a sudden, focal neurological deficit of presumed vascular origin with transient symptoms (<24 hours).[25] Only patients with ischaemic stroke and TIA are eligible for inclusion.
2. Signed informed consent.

### Exclusion criteria

1. Previously diagnosed AF.
2. Contraindication to OAC treatment.
3. Indication for anticoagulant treatment other than AF.
4. Patients with a pacemaker, implantable cardioverter defibrillator or implantable loop recorder.
5. Patients who, according to the investigator, will not be able to comply with the trial protocol.
6. Previous participation in the AF SPICE trial.

The rationale for the age cut-off of 70 years in AF SPICE is partly due to the higher AF yield in elderly patients, as well as the increased benefit of OAC treatment in elderly patients in the case of AF detection. Notably, 74% of stroke patients in Sweden are aged 70 years or older, making the study population quite representative of future implementation. The recruitment window of 14 days is intended to give a margin for the occurrence of patient transfers between centres for thrombectomy, holidays and uncertainty about eligibility. Patients with a

cardiac implantable device are excluded since their very long-lasting continuous heart rhythm monitoring will make them biased with a lower probability of AF detection on a surface ECG recording. Further, ventricular pacing will make reading the patch ECG difficult due to the lack of pacing annotations.

### Withdrawal of patients

A patient can withdraw from the trial at any time, at the discretion of the patient, or by the investigator if deemed medically indicated. If a participant refrains from completing the extended ECG recording protocol, he/she will remain included in the trial and in the intention-to-screen data analysis. A withdrawal of consent means that no further trial data will be collected; however, already collected data will not be erased.

### Patient and public involvement

Patient or public involvement is planned but not yet in place in this trial.

## Interventions
### Extended ECG investigation

Participants randomised to extended ECG investigation will first undergo the standard 0–2 days of ECG monitoring before receiving a continuous one-lead ECG recording using the BioTel ePatch (https://www.biotel.se). The BioTel ePatch is a class IIa CE-marked Holter ECG recorder using a patch for skin adhesion.

The ECG patch will be applied by healthcare professionals at the stroke unit (visit one) at randomisation.

After completion of the 14-day ECG recording, the device should be returned using a prepaid envelope. Following arrival at BioTel, the ECG data will be uploaded to a web-accessed interpretation service (Cardiologs, https://cardiologs.com/) within one working day of arrival.

> **Box 1 Prognostic ECG findings on extended ECG recording which should be communicated to the local investigator.**
>
> Ventricular tachycardia.
> Excessive ventricular extrasystoles (>5%).
> Pause >3 s.
> AV block types II and III.
> Bradycardia <40/minute.

During visit two, participants not diagnosed with AF will have the second ePatch attached by healthcare professionals in connection to a scheduled follow-up visit or attached by the participant/next of kin. The rationale for the second ePatch recording is to increase AF yield, even though the second recording will reasonably identify fewer cases of AF.

The ECG recordings will be scrutinised by a core ECG reading team led by JE. All events in the ECG readings will be manually overread.

### Procedures for handling arrhythmias detected on extended ECG recordings

AF is defined according to the current definition of the ESC AF guidelines,[20] that is, at least 30 s with irregular R-R intervals, the absence of distinct repeating p-waves and irregular atrial activations. When this is met, the local investigator will be notified within one working day by phone and email by the ECG reading team.

If a participant is diagnosed with other arrhythmias with a prognostic implication as defined in box 1, the local investigator will be notified by email within three working days. However, due to the high prevalence of ventricular arrhythmias during 14 days of continuous ECG in this population, a selection based on team discussion is made for ventricular tachycardia. If more rapid action is deemed necessary by the ECG reading team, appropriate measures will be undertaken. When no significant arrhythmias are noted, the local investigator will be notified within 1 week of the ECG upload.

### Procedures for poor signal quality and short recordings in extended ECG

If the signal quality of the extended ECG recording is deemed to be insufficient, the participant should be offered another patch ECG recording. If this second recording has insufficient signal quality or if it is not possible to arrange for another patch ECG recording, a traditional Holter recording or an ECG event recording could be used. However, it is important for the power of the trial to achieve as long ECG recordings as possible in the intervention group.

At least 7 days of ECG recording should be yielded from the ECG patch on visit one. If less than 7 days are recorded, another recording should be offered. The same minimum ECG duration of 7 days applies for ECG recording at visit two.

If a participant randomised to the intervention arm with a negative ECG recording (ie, no AF detected) suffers a recurrent ischaemic stroke within 2 months from visit one, the second ECG patch recording (visit two) may be started earlier than 2 months, to the discretion of the local investigator.

### OAC initiation following AF detection in extended ECG recording

Patients diagnosed with AF during the long-term ECG recordings initiated in connection with randomisation should be informed by the local investigator and offered treatment with OAC if there are no contraindications, based on the stroke risk of at least $CHA_2DS_2$-VASc three points in randomised patients.

Treatment with OAC and its long-term follow-up will be handled within standard care and according to national guidelines.

When indicated, OAC treatment should be initiated within five working days from the reception of long-term ECG results. Long-term follow-up of OAC treatment will be done using the national drug prescription registry.

All other treatments and investigations will be at the discretion of the local investigator.

### Standard care

Participants randomised to standard care should undergo 48±24 hours of continuous ECG monitoring, using inpatient ECG telemetry monitoring or Holter recording. The duration of the monitoring should be recorded by the site and noted in the electronic case report form (REDCap, Vanderbilt University, TN, USA).

The trial will not collect any data from the standard recordings apart from its duration and containment of AF. ECG recordings of newly diagnosed AF in the standard care arm will be collected and stored in the trial.

## Outcomes

### Primary endpoint

Difference between the two arms in a combined endpoint of recurrent ischaemic stroke, intracerebral bleeding and all-cause mortality during at least 36 months of follow-up.

### Secondary endpoints

- ► Individual components of the primary endpoint.
- ► Major bleeding, including intracranial bleeding.
- ► Myocardial infarction.
- ► Pacemaker implantation.
- ► Anticoagulation treatment.
- ► AF prevalence.
- ► Adherence to long-term ECG recording.
- ► Incidence of other prognostic arrhythmias during long-term ECG recording.

### Participant timeline: intervention arm

Participants will be screened for consent during admission for ischaemic stroke or TIA, within 14 days of admission. ECG investigation of 0–48 hours prior to randomisation is allowed. Extended ECG recording is initiated during admission for ischaemic stroke/TIA, and the participant

will return the device via mail. The second extended ECG recording should be initiated at the follow-up visit or by mailing the device to the participants for self-application. Following completion of the second 14-day ECG recording, no further personal visits are made in the trial.

### Participant timeline: standard arm
Starting or completing the ECG recording in the standard arm following discharge is allowed. Following completion of the ECG investigation initiated in the standard arm, no further personal visits are made in the trial.

### Sample size
The sample size was calculated based on the primary outcome.

There were very few studies available with outcome data matching the age and morbidity of the AF SPICE trial. Very few studies on patients with an index stroke incident were available; in particular, there was a lack of data regarding patients with recent strokes and AF without OAC treatment. For this, we have made estimations and extrapolations using the very few available studies on patients with AF without OAC treatment. In particular, extrapolations have been made with regard to age since most available studies have included patients considerably younger than what is expected in AF SPICE. A list of studies used in sample size estimations is available in online supplemental appendix.

For the estimations of AF detection proportions, available studies on short-term (48±24 hours) ECG recordings following stroke and corresponding studies on long-term (14 days) recordings were used. As with the outcome data, no studies with similar populations and ECG recording durations were available. Furthermore, the AF detection/incidence was estimated for the entire follow-up for both arms, reflecting the additional detection within standard care after the index event, thereby reflecting the actual risk exposure in both arms during follow-up. A list of the AF detection studies available for this estimation is reported in online supplemental appendix.

**Table 2** Event rates and AF detection rates used in sample size calculation

| | |
|---|---|
| Endpoint* risk in patients with AF and OAC treatment | 7.3% |
| Endpoint risk in patients with AF and no OAC treatment | 17% |
| Endpoint risk in patients without AF | 5% |
| AF detection proportion in standard arm† | 6% |
| AF detection proportion in intervention arm† | 19% |

*Endpoint including recurrent stroke, all-cause mortality and intracerebral bleeding.
†Including AF detection during follow-up.
AF, atrial fibrillation; OAC, oral anticoagulation.

The assumptions for the sample size calculation are reported in table 2. The complete sample size calculation is provided in the appendix. For the yearly risk of the composite endpoint, we estimate 17% in patients with AF and no OAC treatment, 7.3% in patients with AF and OAC treatment and 5% risk in patients without AF. We further estimate an AF detection proportion of 6% in the control group and 19% after two ECG recordings in the intervention group, and 13% with untreated AF in the control group, resulting in a sample size of 2718 for 3 years of fixed follow-up time after inclusion. Adding 20% for participant dropouts and treatment discontinuation, a total sample size of 3262 is needed, corresponding to 1631 in each arm based on 3 years of follow-up, 80% power and a double-sided significance level of 5%. The study is powered to detect a 9% reduction in the composite endpoint in the group with extended ECG screening.

### Recruitment
Patients will be screened and included during ischaemic stroke or TIA admissions in stroke units. Participating centres report at least 200 stroke/TIA admissions yearly. With the anticipated participation of 30 sites, they need to randomise 0.6 patients weekly during the 3 years of inclusion to fulfil the recruitment target of 3262 participants.

### Allocation
Participants will be randomly assigned to either a control or intervention group with a 1:1 allocation as per a computer-generated randomisation schedule stratified by site in the online REDCap database. Randomisation will be performed immediately following the collection of the necessary baseline data.

### Blinding
Due to the nature of the intervention, neither participants nor staff will be blinded to allocation.

### Data collection methods: primary outcome
Primary endpoint variables are collected from Swedish healthcare registers as specified in table 3.

### Data collection methods: secondary and exploratory outcomes
Variables collected for secondary and exploratory outcomes are reported in table 4.

### Baseline data entered at the inclusion of all participants
Data entered at baseline for all participants are listed in the online supplemental table. The remaining baseline variables will be collected from the Riksstroke register.[5]

### Baseline data entered in the control and intervention arms
Data entered at baseline in the standard and intervention arms are listed in the online supplemental appendix.

### Baseline register data from the Riksstroke register
Baseline data from Riksstroke are listed in the online supplemental appendix.

**Table 3** Primary outcomes

| Variable | Coding International Classification of Diseases 10th Revision | Source | Validity |
|---|---|---|---|
| Ischaemic stroke | I63 | Riksstroke | High |
| Stroke not specified | I64 | Riksstroke | High |
| All-cause mortality | N/A | National register of death | Very high |
| Intracerebral bleeding (spontaneous) | I61 | Riksstroke | High |

## ECG data

There are different ECG variables for the control and intervention arms (online supplemental appendix).

1. In the control arm, the local investigator or delegate will report if ECG was recorded using ECG telemetry and/or Holter recording, the duration of each recording and whether AF was diagnosed.
2. In the intervention arm, ECG data from patch recordings will be stored in the patch hardware and returned by the participant by mail. Data from ECG patch hardware will be uploaded by the patch provider into the Cardiologs database. ECG patch data will be scrutinised by the core reading team, which also enters ECG data manually into the REDCap database.

For the second ECG patch recording initiated 2–4 months following randomisation, data will be collected on whether the patch is applied within healthcare or by the participant.

## Cost-effectiveness

The cost-effectiveness analysis will be based on a Markov cohort model where the prevalence of AF, use of OACs, clinical events and all-cause mortality will be collected from the AF SPICE study. The cost of clinical events, age-specific utilities and stroke deaths will be collected from the literature. The number of gained life years, quality-adjusted life years and cost of the screening process will be calculated.

## Retention

At the majority of included sites, the second visit for the intervention group coincides with an outpatient follow-up visit after the stroke, which will give the participant motivation to attend the visit. Patients receive no reimbursement for participation.

## Statistical methods

The main analysis of the outcome will be based on time to event (endpoint) and performed according to the intention to treat principle. The cumulative incidence of the combined endpoint and its individual components will be presented for each treatment group (standard and extended ECG).

For the primary analysis of the combined endpoint, the log-rank test will be used to test for treatment group differences. Cox proportional hazard regression will be

**Table 4** Secondary outcomes

| Variable | Coding International Classification of Diseases 10th Revision | | Source |
|---|---|---|---|
| Myocardial infarction | I21, I22, I25.2 | | National inpatient register, Riksstroke |
| Major bleeding | Gastrointestinal bleeding | K226, K250, K252, K254, K256, K260, K262, K264, K266, K270, K272, K274, K276, K280, K282, K284, K286, K290, K625, K661, K920, K921, K922, I850 and I983 | |
| | Urogenital bleeding | N02, R319, N939, N950 and N501A | |
| | Other bleeding | H113, H313, H356, H431, H450, H922, I312, J942, M250, R04, R58, T810, D500, D629 and T792 | |
| | Subdural, epidural and subarachnoid haemorrhages (including traumatic) | I60, I61, I62, S064, S065 and S066 | |
| | Procedure codes for transfusion | DR029, DR033 and Z513 | |
| Atrial fibrillation/flutter | I48 | | |
| Pacemaker implantation | FPE00, FPE10, FPE20 and FPE26 | | |
| Oral anticoagulation and antiplatelet treatment, non-steroidal anti-inflammatory drug | B01A, N02BA and M01A | | National drug dispense register |

used to estimate the hazard ratios with 95% CIs and to analyse associations with AF incidence and/or effect modification by age, comorbidity, biometrics and supraventricular activity.

The secondary analysis of each individual component of the primary endpoint will account for competing risks. We will use cause-specific models (where the rate of each individual endpoint is based on patients who have not yet experienced any of the endpoints) and subdistributional hazard models (where the rate is based on patients who have not yet experienced the endpoint under study). Other analysis, including baseline characteristics, will be presented by proportions (binary variables) and means (continuous variables) with 95% CIs in each ECG group. Corresponding group comparisons will be carried out by $\chi^2$ tests and t-tests. An outcome with a p value <0.05 will be considered statistically significant.

### Data monitoring committee

An independent Data Safety and Monitoring Board (DSMB) will be appointed at the start of the main trial, that is, after the pilot phase. The DSMB will have a consulting role on the steering committee. The DSMB will monitor and annually provide statements on:

1. Cumulative inclusion rate, balance in baseline characteristics, duration and yield of ECG recordings during the inclusion phase.
2. Event rate in the standard and intervention groups annually.

### Harms

All diagnostic tools used in the trial are approved by regulatory bodies. Any adverse events noted in connection with investigations or treatment should be reported to authorities according to practice.

### Auditing

A monitoring plan for the trial has been established, with the following areas:

1. Investigator qualifications and delegation log.
2. Written consent forms, accuracy and completeness.
3. Subject eligibility.
4. Source data, accuracy and completeness.
5. Case report form entries in REDCap, unverified and incomplete entries by site.
6. Independent readers will scrutinise 5–10% of ECG reports in the intervention arm.

## ETHICS AND DISSEMINATION
### Independent ethics committee

Ethical approval has been provided by the Swedish Ethical Review Authority (Reference 2021–02770). Before entering the trial, verbal and written information is given to the patient and an informed consent form is signed. The trial will be conducted according to the ethical principles of the Declaration of Helsinki[26] and national regulatory standards.

### Ethical considerations

The optimal duration of the ECG investigation following an ischaemic stroke/TIA is not known.[20]

The ethical issues in this trial are partly connected to the knowledge gap about ECG monitoring duration and the clinical practice that has evolved. Several stroke units are applying ECG monitoring protocols with longer durations than the recommended standard. Randomising patients to a monitoring protocol with a shorter ECG screening duration may be regarded as lowering investigation ambitions. Since the long-term efficacy of different ECG monitoring strategies is unknown, the randomisation to one of the arms in this trial is considered ethically balanced against the option of continuing to use monitoring strategies with unknown efficacy and unknown cost-effectiveness.

### Protocol amendments

Important modifications to the protocol will be summarised consecutively in the protocol and disseminated to investigators. Major protocol changes, that is, changes in eligibility criteria or patient information texts, are subjected to ethical review.

### Consent

Trained research nurses or investigators will introduce the trial. Patients will receive written information. Research nurses or investigators will discuss the trial in light of the written information given, and patients will have an opportunity to discuss the information and ask questions.

In patients with aphasia or an inability to write following a stroke, it seems reasonable to collect informed consent from the next of kin, that is, a spouse or other close relative, if some other mode of communication is established between the patient and the next of kin.

### Confidentiality

At the trial sites, all log lists containing personal identification data will be stored securely. All trial data will be stored in a REDCap database, accessible by multifactor authentication. All actions in the REDCap database are logged. Data in the ECG reading system, Cardiologs, are pseudonymised and accessible with a password.

#### Author affiliations
[1]Department of Clinical Sciences, Danderyd Hospital, Karolinska Institutet, Stockholm, Stockholm, Sweden
[2]Department of Medicine, Huddinge, Karolinska University Hospital, Karolinska Institutet, Stockholm, Stockholm, Sweden
[3]Section of Neurology, Department of Clinical Sciences, Lund University, Lund, Sweden
[4]School of Medicine, Department of Neurology, Orebro universitet, Orebro, Örebro, Sweden
[5]Department of Medicine, Alingsas lasarett, Alingsas, Sweden
[6]Department of Clinical Sciences, South Hospital, Karolinska Institutet, Stockholm, Stockholm, Sweden
[7]Department of Public Health and Clinical Medicine, Umeå University, Umea, Sweden
[8]Department of Statistics, USBE, Umeå University, Umea, Sweden
[9]Department of Medical Sciences, Uppsala University, Uppsala, Sweden

**Acknowledgements** We are grateful to Ida Hed Myrberg and Letizia Orsini at Karolinska Institutet Biostatistics Core Facility for assistance with the power calculation.

**Contributors** JE conceived the trial and drafted the protocol. PW, KS, SÅ, ME and EI made major contributions to trial design. ER, ES, BN, MvE, KH, WG, JOS and SS made contributions to the trial design. All authors approved the final manuscript. AF SPICE is conducted in collaboration with the Swedish Stroke Register, Riksstroke.

**Funding** AF SPICE is funded by grants from the Swedish Research Council, the Swedish Heart & Lung Foundation, the Swedish Stroke Foundation and the Stockholm Region (ALF). The funding sources had no role in the design of this trial and will not have any role during its execution, analyses, interpretation of the data or decision to submit results.

**Competing interests** JE has received consultant or lecture fees from Roche Diagnostics, Pfizer, Bristol Myers Squibb, Boehringer Ingelheim, Piotrode and Philips, as well as research grants from the Swedish Research Council, the Swedish Heart & Lung Foundation, the Swedish Innovation Agency and the Stockholm Region. KS, EI and ER report no disclosures. ES reports consultant or lecture fees from Bayer, Bristol-Myers Squibb, Pfizer, Boehringer-Ingelheim, Johnson & Johnson and Merck Sharp & Dome, as well as research grants from the Swedish Research Council, the Swedish Heart & Lung Foundation, the Åke Wibergs Foundation and CIMED. BN reports Data Safety Monitoring Board fees from Astra Zeneca. MvE is the chairman of Riksstroke, the Swedish Stroke Register. KH, WG, JOS, SS and ME report no disclosures. SÅ reports lecture fees paid to the institution from Bristol-Myers Squibb, Pfizer and Boehringer-Ingelheim and is a member of the steering committee of Riksstroke, the Swedish Stroke Register. PW reports consulting fees, fees as a Clinical Events Committee member and unrestricted grants from Abbott. PW is a member of the steering committee of Riksstroke, the Swedish Stroke Register.

**Patient and public involvement** Patient or public involvement is planned but not yet in place in this trial.

**Patient consent for publication** Consent obtained directly from patient(s).

**Provenance and peer review** Not commissioned; externally peer reviewed.

**ORCID iDs**
Johan Engdahl http://orcid.org/0000-0002-1677-7215
Eva Isaksson http://orcid.org/0000-0003-0289-8750

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
