## [Reviewer comments · BMJ Open]

ARTICLE DETAILS

TITLE (PROVISIONAL)	A multicenter, national, investigator-initiated, randomised, parallel-group, register based superiority trial to compare extended ECG monitoring versus standard ECG monitoring in elderly patients with ischemic stroke or transient ischemic attack and the effect on stroke, death and intracerebral bleeding – The AF SPICE protocol.
AUTHORS	Engdahl, Johan; Straat, Kajsa; Isaksson, Eva; Rooth, Elisabeth; Svennberg, Emma; Norrving, Bo; Euler, Mia; Hellqvist, Kjersti; Gu, Weigang; Ström, Jakob; Själander, Sara; Eriksson, Marie; Åsberg, Signild; Wester, Per

VERSION 1 – REVIEW

REVIEWER	Koh, Keng Tat Sarawak Heart Center
REVIEW RETURNED	18-Apr-2023

GENERAL COMMENTS	Congratulations to the authors for taking up this study. Prolonged ECG monitoring after an ischemic stroke was known to increase the detection of atrial fibrillation, but there is always a gap of evidence on improving the clinical outcomes. I wish the study group all the best in this study.
---

REVIEWER	Kishore, Amit University of Manchester Institute of Cardiovascular Sciences
REVIEW RETURNED	14-May-2023

GENERAL COMMENTS	The authors have attempted to tackle an important aspect in secondary prevention. I offer my congratulations for setting up this study. This is a multi-centre study randomising patch devices for Af detection after IS/TIA Abstract- The abstract should make clear this is a study for secondary prevention, with emphasis on why it is important to screen with wearable technology and what advantages they offer Manuscript- The introduction should again make clear that this is a study looking a secondary prevention-the authors should describe when is recurrent stroke most common, and what early detection of AF might offer. Is there a similar risk for AF detected after stroke, as opposed to AF detected in primary prevention and perhaps the authors could comment on neurocardiogenic effect of stroke.
---

	Definitions- The authors should define how stroke is diagnosed- clinical/radiological, How TIA is diagnosed- Is this based on motor deficits, speech deficit only? How is AF diagnosed and what duration of AF is recorded as significant- for example 30 sec or any duration of AF. The authors should clarify why 14 days was chosen as time for randomisation. A number of studies have shown early AF detection after stroke , within the first week- including Find AF study. Is this largely related to logistical issues? And if so , perhaps this should be mentioned in the limitations. The authors should clarify why 70y was chosen for inclusion? Was there any information from the pilot data that was analysed including patient reported outcome that might have helped with this protocol? It would appear that there was no patient /public involvement before formulating the protocol, which is a concern.
--	---

REVIEWER	Cameron, Alan University of Glasgow
REVIEW RETURNED	25-May-2023

GENERAL COMMENTS	Many thanks for the opportunity to review the protocol for a very interesting study that addresses an important clinical issue. I have a number of suggestions that I hope will help to describe the study more clearly for readers which I have included below: Abstract Line 37 “monitoring” is missing after ECG Line 43 and throughout manuscript “submitted for” – would suggest consider rephrasing as “admitted with” Introduction Line 96 Would suggest “leaves” rather than “leave” Line 97 “...untreated risk factor” – would suggest “undiagnosed” or “undiagnosed and untreated” Would suggest adding data from the EPACS RCT (Eur J Med Res (2019) 24:25) and Tsivgoulis meta-analysis (Neurology 2022;98:e1942-e1952). Methods Line 136 and throughout manuscript It is unclear what is meant by “48 +/-24 hours” – is this 1) a minimum of 48 hours plus/minus a further 24 hours; or 2) 24 or 48 hours. Would suggest clarifying throughout. I have classed the methods as not being described sufficiently in the above responses pending clarification of this point.
--

	Line 139 It is interesting that neither AF nor oral anticoagulation are secondary outcomes. Would suggest including these as secondary outcomes or providing strong justification for not including these as outcomes since they are relevant from both a clinical and scientific perspective. Line 164 As above re: “submitted for...” Line 175 It would be helpful to clarify the definition of TIA and what measures are being taken to avoid including stroke mimics within this group. Line 184 It would be helpful to provide the rationale for excluding people with a pacemaker or ICD. Line 196 It would be helpful to provide the rationale and justification for not including patient or public involvement in the trial. Line 200 It would be helpful to provide the rationale for selecting the BioTel patch rather than other ECG patch(es) that are available. My understanding is that BioTel patches provide AI output which requires review and verification of detected arrhythmias. It would be helpful to clarify this point and describe who will verify detected arrhythmias assuming that is the process. Line 204 It would be helpful to provide details of the timeframe for the ECG patch report being communicated to clinicians. Line 208 The rationale for including a second 14-day ECG patch in the trial should be described. Line 227 “has” rather than “have” Line 248 CHA2DS2-VASc ≥ 3 seems a high cut-off for recommending anticoagulation. ESC guidelines recommend anticoagulation for CHA2DS2-VASc ≥ 2 and suggest considering anticoagulation for CHA2DS2-VASc = 1. Would suggest reconsidering or providing strong justification for the proposed cut-off of CHA2DS2-VASc ≥ 3. Line 256 As above re: 48 +/-24 hours Line 268 As above re: not having AF or anticoagulation as outcomes Line 304
--	---

	Estimated 17.7% annual risk of the composite endpoint for people with AF and no OAC seems very high. It would be helpful to provide justification for this estimate. Line 306 Estimated 19% AF detection seems very high on the basis of AF detection rates from previous studies. The only studies in the appendix table with AF detection rates greater than this are appendix references 23 and 24 which were conducted in people with cryptogenic stroke rather than unselected stroke and included a very small number of participants (20 and 62 respectively). It would be helpful to justify the estimated 19% AF detection. The sample size seems mainly based on anticipated AF detection and it would be helpful to better describe the contribution of rates for ICH and mortality to the sample size calculation. Line 314 As above re: TIA definition Line 325 Will the analysis be blinded as this seems feasible to do? Line 361 Will randomisation be stratified? Line 404 Why will only 5-10% of ECGs be scrutinised and what will the scrutinization involve? Line 436 Overall The proposed methodology for cost-effectiveness analysis is not described. I have classed the outcomes as not sufficiently described in the above bullet points pending clarification of this point. Table 1 This includes a mixture of recommendations for people with unselected stroke subtypes and people with cryptogenic stroke. It would be helpful to split the table into recommendations for these two groups. Table 2 It would be helpful to describe the justification for communicating 1) all episodes of VT, 2) excessive ventricular extrasystoles and 3) Mobitz type 1 second degree AV block as this will involve a lot of work for local investigators to review. The clinician time to review these findings should also be factored into cost-effectiveness analyses. Figure 1 As per some of the comments above it would be helpful to clarify: 1. If all participants (i.e. including those randomised to 14+14 days) will initially have 0-2 days of ECG recording. It appears this way in the figure but not in the text. This is crucial as it will
--	--

	determine whether the population is a cohort of people with “unselected” or “cryptogenic” stroke. 2. The justification for the 2-4 month interval for fitting the second ECG patch. 3. The justification for including a second 14 day ECG patch. Many thanks.
--	--

VERSION 1 – AUTHOR RESPONSE

Reviewer: 1

Dr. Keng Tat Koh, Sarawak Heart Center

Comments to the Author:

Congratulations to the authors for taking up this study. Prolonged ECG monitoring after an ischemic stroke was known to increase the detection of atrial fibrillation, but there is always a gap of evidence on improving the clinical outcomes. I wish the study group all the best in this study.

Thank you for this comment!

Reviewer: 2

Dr. Amit Kishore, University of Manchester Institute of Cardiovascular Sciences

Comments to the Author:

The authors have attempted to tackle an important aspect in secondary prevention. I offer my congratulations for setting up this study. This is a multi-centre study randomising patch devices for Af detection after IS/TIA

1. Abstract- The abstract should make clear this is a study for secondary prevention, with emphasis on why it is important to screen with wearable technology and what advantages they offer.

The secondary prophylactic nature of the ECG investigation post ischemic stroke/TIA and the advantages of the ECG patch have been added into abstract:

“Atrial fibrillation (AF) is a major risk factor for ischemic stroke and transient ischemic attack (TIA), and AF detection can be challenged by asymptomatic and paroxysmal presentation. Long-term ECG after ischemic stroke or TIA is recommended by all major societies in cardiology and cerebrovascular medicine as a secondary prophylactic measure.”

“In the intervention arm, patients will undergo 14+14 days of continuous ECG monitoring three months apart using an ECG patch device, which will provide an easy-accessed, well-tolerated 14-days continuous ECG recording.”

2. Manuscript- The introduction should again make clear that this is a study looking a secondary prevention.

First bullet of “Strengths and limitations” revised:

- “First register-based randomised clinical trial on secondary preventive ECG screening post stroke/TIA with stroke and all-cause mortality endpoints”

Last sentence of Introduction revised:

“With this background, there is an urgent need for a randomised trial to investigate the efficacy of secondary prophylactic AF screening following ischemic stroke.”

3. the authors should describe when is recurrent stroke most common, and what early detection of AF might offer. Is there a similar risk for AF detected after stroke, as opposed to AF detected in primary prevention and perhaps the authors could comment on neurocardiogenic effect of stroke.

One-year rate of ischemic stroke recurrence has been reported to be 3-5% (Andersen Stroke 2015; Khanevski Acta Neurologica Scandinavica 2019) with the steepest rate during the first 3-6 months following the index event. Five-year recurrence rates are reported at 12% from the same two publications. Recurrence risk seem more linear following the first year after the index event. However, we assume that patients with cardioembolic stroke from atrial fibrillation will have higher risk for recurrence in the case of non-detection of AF and non-treatment with oral anticoagulants. The stroke risk in the general population is apparently lower, as reported from the STROKESTOP and LOOP trials (Svennberg The Lancet 2021; Svendsen the Lancet 2021) where a yearly risk of 1% is reported from 75-year-old populations during long-term follow-up.

In primary prevention AF screening, there is obviously no clinical index event related to the screening procedure, and the populations studied in primary preventive AF screening generally show a much lower morbidity/comorbidity, (Svennberg, the Lancet 2021; Perez, NEJM 2019) making a comparison difficult. However, with the use of implantable cardiac monitors, the AF yield in secondary preventive studies (Sanna NEJM 2014; Buck JAMA 2021; Bernstein JAMA 2021; Triantafyllou Ann Neurol 2020) are similar to the AF yield reported from primary preventive trials (Svendsen the Lancet 2021; Healey Circulation 2017)

The following was added to the third paragraph of Introduction:

“The risk for stroke recurrence is highest during the first 3-6 months following admission, and the one-year recurrence risk has been reported to be 3-5%{Andersen, 2015 #2432;Khanevski, 2019 #2437}.”

Neurologic influence on heart activity following stroke could have a wide variety of implications such as ECG morphology changes, Troponin T elevation and arrhythmias, including atrial fibrillation and sudden death (Sposato, JACC 2020). However, there is no evidence on how we confidently could differ post-stroke arrhythmias from more permanent forms. Hence, atrial fibrillation detected post stroke will be treated as a stroke risk factor.

4. Definitions- The authors should define how stroke is diagnosed- clinical/radiological, How TIA is diagnosed- Is this based on motor deficits, speech deficit only?

The diagnosis of TIA is not restricted to motor deficits and/or speech deficit only. The diagnostic definitions of stroke and TIA adhere to the definitions used in the Swedish Stroke Register (Riksstroke) from which data on recurrent cerebrovascular events is collected for the study. The definitions also concur with the classical WHO definition of stroke and TIA (Aho Bull World Health Organ 1980). The diagnosis of TIA at Swedish stroke units has been validated in Riksstroke where a

high rate of interobserver agreement on TIA diagnoses were found (Buchwald, Neuroepidemiology 2015).

The definitions of stroke and TIA were added to Eligibility Criteria:

“Stroke is clinically defined as an acute onset of focal neurological deficit of presumed vascular origin lasting for ≥ 24 hours or resulting in death. Stroke is further categorized as ischemic, or as intracerebral or subarachnoid haemorrhage based on computed tomography (CT) and/or magnetic resonance imaging (MRI) of the brain. Transient ischemic attack (TIA) is defined as a sudden, focal neurological deficit of presumed vascular origin with transient symptoms (< 24 hours). (ref Aho et al 1980) Only patients with ischemic stroke and TIA are eligible for inclusion.”

Inclusion Criteria was clarified:

Patients aged ≥ 70 years admitted to a stroke unit with a diagnosis of ischemic stroke or TIA within 14 days from inclusion.

5. How is AF diagnosed and what duration of AF is recorded as significant- for example 30 sec or any duration of AF.

The definition of AF (according to the ESC) is added to the “Procedures for handling arrhythmias detected on extended ECG recording”:

“Atrial fibrillation is defined according to the current definition of the ESC AF guidelines¹⁷, i.e. at least 30 seconds with irregularly irregular R-R intervals, absence of distinct repeating p-waves and irregular atrial activations. When this is met, the local investigator will be notified within one working day by phone and email by the ECG reading team.”

6. The authors should clarify why 14 days was chosen as time for randomisation. A number of studies have shown early AF detection after stroke, within the first week- including Find AF study. Is this largely related to logistical issues? And if so, perhaps this should be mentioned in the limitations.

Thank you for bringing this uncertainty to our attention. The intention is to include and randomise patients as soon as possible after admission to the stroke unit, and preliminary data after the first 700 randomisation show that participants are randomised 3,3 (range 1-13) days following admission. The occurrence of patient transfer between centres for thrombectomy, holidays/weekends and uncertainty about eligibility could delay consent and randomisation. Considering this, 14 days seem reasonable and reflecting the typical time window for stroke admissions, with a slight margin. Previous hallmark studies in AF detection post stroke and TIA, such as the EMBRACE and CRYSTAL AF allowed recruitment windows of 6 and 3 months respectively, both restricted to cryptogenic stroke. Hence, AF SPICE is given a pragmatic design with broad inclusion criteria and a recruitment window adopted for clinical practice.

We have elaborated the rationale for the recruitment window in “Eligibility criteria”:

“The recruitment window of 14 days is intended to give a margin for the occurrence of patient transfer between centres for thrombectomy, holidays and uncertainty about eligibility.”

7. The authors should clarify why 70y was chosen for inclusion?

Thank you for addressing this important issue and giving us an opportunity to clarify. So far, majority of studies on AF detection post stroke/TIA have included relatively young stroke populations, particularly studies using implantable cardiac monitors (ICM). For instance, in the CRYSTAL-AF,

PROACTIA, PER DIEM and STROKE-AF trials, the median age of the participants were 62, 69, 64 and 67 years respectively. In the MonDAFIS trial, using surface ECG recordings, mean was 66 years. The mean age for patients admitted for stroke in Sweden was 75 years (2021 data).

The rationale for the age cut-off of 70 years in AF SPICE is partly due to the higher AF yield in elderly patients, as well as the increased benefit for oral anticoagulation treatment in elderly patient in the case of AF detection. Notably, 74% of stroke patients in Sweden are aged 70 years of above, making the study population quite representative for future implementation.

We have clarified the selected age cut-off in “Eligibility criteria”:

“The rationale for the age cut-off of 70 years in AF SPICE is partly due to the higher AF yield in elderly patients, as well as the increased benefit for oral anticoagulation treatment in elderly patient in the case of AF detection. Notably, 74% of stroke patients in Sweden are aged 70 years of above, making the study population quite representative for future implementation. “

8. Was there any information from the pilot data that was analysed including patient reported outcome that might have helped with this protocol? It would appear that there was no patient/public involvement before formulating the protocol, which is a concern.

Unfortunately, we were not able to arrange public/patient involvement during the pilot phase. The pilot study brought mainly information on inclusion rates and data collection protocols. However, patient involvement in study management is intended for the remainder of the study. We revised Patient and Public involvement:

Patient or public involvement is planned but not yet in place in this trial.

Reviewer: 3

Dr. Alan Cameron, University of Glasgow

Comments to the Author:

Many thanks for the opportunity to review the protocol for a very interesting study that addresses an important clinical issue. I have a number of suggestions that I hope will help to describe the study more clearly for readers which I have included below:

1. Abstract Line 37 “monitoring” is missing after ECG

This is revised accordingly.

2. Line 43 and throughout manuscript “submitted for” – would suggest consider rephrasing as “admitted with”

Thank you for this suggestion, this has been revised.

3. Introduction Line 96 Would suggest “leaves” rather than “leave”

This is revised accordingly.

4. Line 97 “...untreated risk factor” – would suggest “undiagnosed” or “undiagnosed and untreated”

We have revised into “undiagnosed and untreated”.

5. Would suggest adding data from the EPACS RCT (Eur J Med Res (2019) 24:25) and Tsigoulis meta-analysis (Neurology 2022;98:e1942-e1952).

Thank you for the recommendation of the EPACS trial. The AF detection rate along with the age of the study population is of interest for our study and it will be added to appendix table 5.

Regarding the Tsigoulis reference, the RCTs included in this meta-analysis (except for the EMBRACE trial) are studies using Implantable Cardiac Monitoring (ICM), why we prefer to keep the Schnabel reference for ECG monitoring post stroke in general, although being a slightly older reference.

6. Methods Line 136 and throughout manuscript It is unclear what is meant by “48 +/-24 hours” – is this 1) a minimum of 48 hours plus/minus a further 24 hours; or 2) 24 or 48 hours. Would suggest clarifying throughout. I have classed the methods as not being described sufficiently in the above responses pending clarification of this point.

Thanks for bringing this to our attention, this has been clarified throughout the manuscript:

(48 ±24 hours i.e. range 24-72 hours)

7. Line 139 It is interesting that neither AF nor oral anticoagulation are secondary outcomes. Would suggest including these as secondary outcomes or providing strong justification for not including these as outcomes since they are relevant from both a clinical and scientific perspective.

Thank you for bringing this to our attention. AF prevalence and anticoagulation treatment are secondary outcomes of the AF SPICE study as noted in our trial registration, but they were not listed in the manuscript out of inadvertence. This is now corrected in Secondary Endpoints:

“Secondary endpoints

- Individual components of the primary endpoint
- Major bleeding including Intracranial bleeding
- Myocardial infarction
- Pacemaker implantation
- Anticoagulation treatment
- Atrial fibrillation prevalence
- Adherence to long-term ECG recording
- Incidence of other prognostic arrhythmia during long-term ECG recording”

8. Line 164 As above re: “submitted for...”

This has been revised.

9. Line 175 It would be helpful to clarify the definition of TIA and what measures are being taken to avoid including stroke mimics within this group.

All eligible patients are identified at Stroke Units and registered in The Swedish Stroke Register (Riksstroke) with either ischemic stroke or TIA according to the international classification of diseases, 10th revision (the ICD-10). This procedure diminishes the risk of inclusion of mimics as the diagnosis of TIA at Swedish stroke units has been validated in Riksstroke where a high rate of interobserver agreement on TIA diagnoses were found (Buchwald, Neuroepidemiology 2015).

See also response to Reviewer 2, point 4 above and the revisions made to this similar comment.

10. Line 184 It would be helpful to provide the rationale for excluding people with a pacemaker or ICD.

Thank you for the opportunity to clarify this exclusion criteria. Patients with a cardiac device are excluded for two reasons, firstly because of the continuous and long-lasting (often for many years) heart rhythm monitoring these patients are subjected to. Two plus two weeks of external/surface ECG monitoring have a very low probability of detecting new AF in these patients. Secondly, for patients with ventricular pacing, reading the one-lead patch ECG becomes difficult since there is no annotation of paced beats bringing uncertainty about ventricular activity.

The following was added to Eligibility Criteria:

“Patients with a cardiac implantable device are excluded since their very long-lasting continuous heart rhythm monitoring will make them biased with a lower probability of AF detection on a surface ECG recording. Further, ventricular pacing will make reading of the patch ECG difficult due to the lack of pacing annotations. “

11. Line 196 It would be helpful to provide the rationale and justification for not including patient or public involvement in the trial.

Unfortunately, we were not able to arrange public/patient involvement during the pilot phase. Hence, it was not our intention to actively omit public/patient involvement. Patient involvement in study management is intended for the remainder of the study.

12. Line 200 It would be helpful to provide the rationale for selecting the BioTel patch rather than other ECG patch(es) that are available. My understanding is that BioTel patches provide AI output which requires review and verification of detected arrhythmias. It would be helpful to clarify this point and describe who will verify detected arrhythmias assuming that is the process.

We had some desired specifications regarding the ECG-recording equipment during the set-up of the trial in 2021, one was the capability of recording 14 days of continuous ECG, another was that the ECG recording should be stand-alone without the need for linking to cell phones or other involvement of the patient (apart from the return of the patch). The BioTel epatch was the only patch ECG device available in Sweden fulfilling these specifications at that time.

For ECG reading, BioTel/Philips uses the Cardiologs platform, which is a well-validated, partly AI-based ECG reading service. The output provided for BioTel epatch is very much like a traditional Holter reading, and the BioTel epatch is CE-branded as a Holter device. The web links for these two services are provided in the manuscript.

We have clarified that all events in the ECG readings are manually studied in Extended ECG Investigation:

The ECG recordings will be scrutinised by a core ECG reading team led by JE. All events in the ECG readings will be manually overread.

13. Line 204 It would be helpful to provide details of the timeframe for the ECG patch report being communicated to clinicians.

There are two timeframes for communicating with the local clinician on pathological ECG results. When AF or AFL is detected, the local clinician should be contacted within one working day using both phone and email. For the other significant arrhythmias, the group of the local investigator should be contacted using email within 3 working days.

If no significant arrhythmias are noted, the local investigator will be notified within one week from the upload of the ECG recording. The latter has been clarified in Extended ECG Investigation:

When no significant arrhythmias are noted, the local investigator will be notified within one week from ECG upload.

14. Line 208 The rationale for including a second 14-day ECG patch in the trial should be described.

Thank you for bringing this important point to attention for clarification. Due to the unpredictably random appearance of paroxysmal AF, repeated ECG recordings have higher AF yield than one continuous recording of equal total duration (Diederichsen Circulation 2020), repeated continuous ECG recordings have proven to increase AF yield in the post-stroke setting (Wachter Lancet Neurology 2017), however with a lower yield for the second recording, and even lower for a third recording. Yearly repeated recordings are also applied in the ongoing German FindAF2 study (Start Page | Find-AF2). The AF yield from the two epatch recordings will be reported in the secondary outcome of AF prevalence.

This rationale has been clarified in Extended ECG Investigation:

“The rationale for the second ePatch recording is to increase AF yield, even though the second recording will reasonably identify fewer cases of AF.”

15. Line 227 “has” rather than “have”

Corrected.

16. Line 248 CHA2DS2-VASc ≥ 3 seems a high cut-off for recommending anticoagulation. ESC guidelines recommend anticoagulation for CHA2DS2-VASc ≥ 2 and suggest considering anticoagulation for CHA2DS2-VASc = 1. Would suggest reconsidering or providing strong justification for the proposed cut-off of CHA2DS2-VASc ≥ 3 .

Maybe a misunderstanding, this is not a proposed cut-off, we just recognise that all randomised patients diagnosed with AF in the trial have at least 3 points according to CHADS-VASc, i.e. a strong indication for anticoagulation treatment, particularly with age and previous stroke/TIA as risk factors.

17. Line 256 As above re: 48 +/-24 hours

Revised according to comment #6.

18. Line 268 As above re: not having AF or anticoagulation as outcomes

Revised according to comment #7

19. Line 304 Estimated 17.7% annual risk of the composite endpoint for people with AF and no OAC seems very high. It would be helpful to provide justification for this estimate.

Thank you for bringing this to our attention. Contemporary data on outcomes in stroke/TIA patients with AF but no oral anticoagulation is not available, and we have had to make estimations based on a few sources.

We mainly base the risk on the data reported from the AVERROES trial (Diener, Lancet Neurology 2012), but there are several factors making these data less suitable for our purpose.

Firstly, the AVERROES study recruited AF patients, not stroke patients, meaning that the inclusion of patients in the AVERROES trial is not made at a stroke event, which in turn will lower the risk for recurrent stroke.

Secondly, the duration between stroke/TIA and randomisation is not reported in the AVERROES substudy, bringing the assumption that recurrent stroke risk will be lower in those patients included more than 6 months after stroke/TIA.

Thirdly, the population in this AVERROES substudy is 3-4 years younger than the current mean age in AF SPICE participants, giving a lower risk of recurrent stroke as well. Adding the outcome rates of ischemic stroke, all-cause mortality and intracerebral bleeding from the AVERROES substudy (7,4%+7,9%+1,6%) minus the estimated 30% overlap caused by lethal stroke equals an estimated yearly total risk for our composite endpoint of 14%, to which we add 20% risk increase in AF SPICE participants due to their index stroke event and higher age, resulting in an estimated yearly risk of 17-18% for the composite endpoint.

Outcome data from this study was unfortunately mistyped into the appendix table "Trials in event rates power calculation" and has been corrected:

Yearly outcome rates: Ischemic and unspecified stroke 8,26%; All-cause mortality 7,92%; intracerebral bleeding 1,56%.

Another source contributing to this risk estimation is the EAFT study (Anonymous, Lancet 1993) which randomised patients aged above 25 years with recent TIA or minor stroke and AF to anticoagulation treatment, aspirin or placebo. The Mean age was 73 years and the annual event rate 17% in the placebo group using a composite endpoint of death from vascular disease, any stroke, myocardial infarction or systemic embolism. Vascular deaths and nonfatal strokes were the dominating events during follow-up. Although conducted in a different era, this study shows the magnitude of risk in "untreated" minor stroke/TIA patients with atrial fibrillation. The lower age of the participants in the EAFT trial and the inclusion criteria of minor stroke or TIA imply that patients relevant for the AF SPICE would have even higher risk under the same conditions.

20. Line 306 Estimated 19% AF detection seems very high on the basis of AF detection rates from previous studies. The only studies in the appendix table with AF detection rates greater than this are appendix references 23 and 24 which were conducted in people with cryptogenic stroke rather than unselected stroke and included a very small number of participants (20 and 62 respectively). It would be helpful to justify the estimated 19% AF detection. The sample size seems mainly based on anticipated AF detection and it would be helpful to better describe the contribution of rates for ICH and mortality to the sample size calculation.

As already commented on, AF detection data in the AF SPICE age group are scarce. Adding to that, we randomise patients regardless of stroke etiology, making it even more difficult to find a comparative population. The EMBRACE study (Gladstone NEJM 2014) reported 12% AF yield after two weeks of continuous ECG monitoring, however in patients with cryptogenic stroke and slightly younger than in AF SPICE.

In the FIND AF – randomised study (Wachter Lancet Neurology 2017), 10 days of Holter was used repeatedly and only patients with stroke were included. The first Holter recording yielded 9% AF, and the second 4 percent. Once again, it is worth mentioning that patients randomised in AF SPICE are a few years older than patients included in EMBRACE and FIND AF – randomised. On the other hand, one could speculate that the randomisation of TIA patients will decrease AF yield slightly as compared to stroke patients.

The AF prevalence used for the power calculation in AF SPICE is the prevalence estimated during follow-up, which also explains the rather high proportion of AF of 6% estimated in the control group. Hence, we expect the AF prevalence to increase with 2-4 percent during follow-up in both intervention and control groups.

We can reveal preliminary data from the first 750 randomisations: AF yield after the first epatch is 10% and additional 4% after the second epatch, and the AF yield in the control group is 2%. Hence, these preliminary data are very close to our estimations.

We have added a clarification to Sample Size:

“The estimated AF detection in the control and intervention groups include cases detected during follow-up.”

21. Line 314 As above re: TIA definition

Please see response to comment 9.

22. Line 325 Will the analysis be blinded as this seems feasible to do?

Outcomes will be collected as ICD 10 codes from Swedish health care registers without blinding and will be analysed without any further adjudication since the validity of these registers are high or very high. This is stated in Strengths and Limitations:

“Endpoints from health care registers will not be further adjudicated”

ICD 10 coding of outcome data are reported in tables 4a/4b.

23. Line 361 Will randomisation be stratified?

Randomisation is stratified by site, this is mentioned in Allocation:

“Participants will be randomly assigned to either control or intervention group with a 1:1 allocation as per a computer-generated randomisation schedule stratified by site in the online REDCap database.”

24. Line 404 Why will only 5-10% of ECGs be scrutinised and what will the scrutinization involve?

All epatch ECGs will be read according to comment 12 and 13, the mentioned proportion of 5-10% concerns monitoring/auditing. During epatch ECG monitoring, an independent reader will make a full reading and interpretation of the recording according to appendix table “ECG Data collected in intervention group” and compare with the result from the ECG reading team.

25. Line 436 Overall The proposed methodology for cost-effectiveness analysis is not described. I have classed the outcomes as not sufficiently described in the above bullet points pending clarification of this point.

An overview of the cost effectiveness methodology has been added in Data Collection:

“Cost Effectiveness

The cost-effectiveness analysis will be based on a Markov cohort model where prevalence of AF, use of oral anticoagulants, clinical events and all-cause mortality will be collected from the AF SPICE study. The cost for clinical events, age-specific utilities and stroke death will be collected from the literature. Number of gained life-years, quality-adjusted life years and cost for the screening process will be calculated.“

26. Table 1 This includes a mixture of recommendations for people with unselected stroke subtypes and people with cryptogenic stroke. It would be helpful to split the table into recommendations for these two groups.

Admittedly, there are many different concepts used in the society recommendations, such as ESUS; cryptogenic stroke; stroke and TIA of undetermined origin and non-lacunar ESUS. We have revised the table by inserting verbatim wordings and highlighting when subgroups of stroke patients are targeted.

27. Table 2 It would be helpful to describe the justification for communicating 1) all episodes of VT, 2) excessive ventricular extrasystoles and 3) Mobitz type 1 second degree AV block as this will involve a lot of work for local investigators to review. The clinician time to review these findings should also be factored into cost-effectiveness analyses.

We are happy that you bring this matter into light. A continuous ECG recording of 14 days duration in 75-year-old stroke/TIA patients will return many pathological arrhythmias, and these are more frequent than atrial fibrillation. From the start of the AF SPICE study, the intention was not to report all episodes of VT, there are however difficulties in establishing a single cut-off for VT "burden" being comprehensively applicable. For instance, even a single VT episode could be clinically significant given an extensive duration and high heart rate. Further, a patient could display a large burden of very short VT episodes, giving a clinically significant ventricular arrhythmia burden. On the other hand, in traditional short-term Holter reading, all VT are considered pathological.

However, in the multimorbid and elderly participants included in AF SPICE, we find VT (at least three ventricular beats with a heart rate > 100/min) preliminary in 30-40% of 14-day patch ECGs, making it difficult to consider all cases as clinically significant/relevant.

The pragmatic solution to this has been to report a selection of VT cases to the local investigators. This selection is based on team discussions/decisions in the ECG reading team. As an example, one or two very short VT episodes in an epatch recording are not reported to the local investigator. We are collecting data on the number of communications with local investigators as well as the burden of VTs recorded, making it possible to estimate the workload.

We have added a clarification in Procedures for handling of arrhythmias detected on extended ECG recording:

"However, due to the high prevalence of ventricular arrhythmias during 14 days of continuous ECG in this population, a selection based on team discussion is made for VTs."

28. Figure 1 As per some of the comments above it would be helpful to clarify: A. If all participants (i.e. including those randomised to 14+14 days) will initially have 0-2 days of ECG recording. It appears this way in the figure but not in the text. This is crucial as it will determine whether the population is a cohort of people with "unselected" or "cryptogenic" stroke. B. The justification for the 2-4 month interval for fitting the second ECG patch.

The study design is intended to give low thresholds for participant screening and inclusion, and reflect clinical practice. Because patients with ischemic stroke and TIA are admitted 24/7 to stroke units and immediately attached to ECG monitoring, it would not have been possible to conduct the trial if we wanted "ECG-naïve" patients into randomisation, which is undertaken during office hours.

To the best of our knowledge, there are several criteria for "cryptogenic" stroke, not only duration of ECG monitoring. As stated in inclusion criteria, the study population should consist of unselected ischemic stroke/TIA patients, with the exception of age.

We have added a clarification in Extended ECG Investigation:

"Participants randomised to extended ECG investigation will first undergo the standard 0-2 days of ECG monitoring, before receiving a continuous one-lead ECG-recording using the BioTel ePatch"

Regarding the time window between first and second epatch, there is not much clinical evidence at hand. The interval of 3 months was used in the FIND AF-randomised study (Wachter Lancet Neurology 2017). In a substudy of the LOOP study (Diederichsen Circulation 2020) reported sensitivity for different long-term ECG monitoring options, showing that multiple recordings were more

important than duration for detection of silent AF. However, patient comfort and compliance must be considered as well.

The timing of follow-up visits used by a majority of stroke units in Sweden was a factor that contributed to the decision as well.

29. The justification for including a second 14-day ECG patch.

Please see comment 14.

Many thanks.

Reviewer: 1 Competing interests of Reviewer: none to declare Reviewer: 2 Competing interests of Reviewer: No financial interests I have been a co-applicant in similar grants for secondary prevention involving AF detection and several related systematic reviews. Reviewer: 3 Competing interests of Reviewer: I have an academic interest in AF screening

VERSION 2 – REVIEW

REVIEWER	Cameron, Alan University of Glasgow
REVIEW RETURNED	11-Oct-2023

GENERAL COMMENTS	Many thanks for addressing my previous comments. There are two points that I think it would be helpful to further clarify, which I have summarized below: "19. Line 304 Estimated 17.7% annual risk of the composite endpoint for people with AF and no OAC seems very high. It would be helpful to provide justification for this estimate." Thanks for providing clarification regarding this. It would be helpful to explain this in more detail in the manuscript and discuss the limitations outlined in the response to reviewer comments. I also looked at the data presented in table 3 (8.26% + 7.92% + 1.56%). Could the authors please describe how these result in 17-18% as I do not quite get the same numbers using the approach outlined: 8.26% + 7.92% + 1.56% = 17.74% 17.74 minus 30% = 12.42% 12.42 plus 20% = 15% "20. Line 306 Estimated 19% AF detection seems very high on the basis of AF detection rates from previous studies." Anticipated 19% AF detection does still seem quite high and it would be helpful to the discuss the limitations outlined in the response to reviewer comments in the manuscript. Many thanks
--